# A Dual-Band Polarization-Insensitive Frequency Selective Surface for Electromagnetic Shielding Applications

**DOI:** 10.3390/s24113333

**Published:** 2024-05-23

**Authors:** Muhammad Idrees, Yejun He, Shahid Ullah, Sai-Wai Wong

**Affiliations:** State Key Laboratory of Radio Frequency Heterogeneous Integration, Guangdong Engineering Research Center of Base Station Antennas, Shenzhen Key Laboratory of Antennas and Propagation, College of Electronics and Information Engineering, Shenzhen University, Shenzhen 518060, China; muidrees169@gmail.com (M.I.); heyejun@126.com (Y.H.); shahidkhan@szu.edu.cn (S.U.)

**Keywords:** angularly stability, EM interference, frequency-selective surface, polarization insensitive, shielding effectiveness, spatial filter

## Abstract

This paper presents a novel polarization-insensitive dual-band frequency-selective surface (FSS)-based electromagnetic shield. The miniaturized FSS unit cell consists of a modified Jerusalem crossed loop and a corner-modified square loop. These FSS elements are arranged in a co-planner configuration over a single-layer Rogers 5880 substrate and simultaneously offer effective shielding in the X- and Ku-bands. Moreover, the FSS manifests polarization-independent and angularly stable band-reject filter characteristics over various oblique angles of incidence for both the TE and TM polarizations with virtuous attenuation at both resonances. In addition, the FSS structure is modelled into an equivalent lumped circuit to better analyze the phenomenon of EM wave suppression. A finite prototype of FSS is fabricated and tested. The simulated and measured results are in good agreement, thus making it a potential candidate for RF shielding/isolation applications.

## 1. Introduction

Rapid developments in wireless communication technology have introduced many intelligent devices and communication systems, revolutionizing human life. These radiating devices and communication systems operating in closely spaced frequency bands generate electromagnetic interference. This might cause potential human health hazards and the malfunction or performance failure of other devices, electronic components, and RF systems working nearby. To address this problem, conductive sheets, absorbing surfaces, wire mesh, and metal screens are generally employed; however, they might block all transmissions and be bulky. On the other hand, frequency-selective surfaces (FSSs) are immune to such shortcomings and suitable, where selective shielding is required because of their small size, low cost, stable frequency responses, ease of fabrication, and employability. FSSs are the periodic arrangement of metallic resonant elements in two or three dimensions that can allow/shield/absorb electromagnetic waves and find widespread employability, including, but not limited to, radomes [1], EM absorbers [2,3,4], polarizers [5], antenna gain enhancement [6], RF energy harvesting [7], satellite communication [8], electromagnetic interference mitigation [9,10], and many others.

Several single-layer FSS structures have been studied for various suppression applications recently. In [11], a flexible FSS realized on textile and film substrates using a screen-printed technique is studied to see the effect of the texture printed on each substrate on its shielding characteristics. An FSS shield [12] with an ultrawideband (1.70–15.4 GHz) response is presented. However, it has the drawback of blocking out all communications over a wide range. In [13], a fractal crossed dipole structure is introduced for GSM shielding. This flexible, optically transparent dual-band FSS finds employability in building windows. A miniaturized bandstop FSS realized on a glass substrate in [14] blocks Wi-Fi and WLAN frequencies and can be used for indoor security purposes. However, it exhibits a limited angular sensitivity up to only 45°. In [15], a pair of interdigitated hexapoles and intermediate tripoles is used to prevent transmission at 2.14 GHz and 5.13 GHz, but significant frequency shifts are observed at the upper stopband for the TM mode up to 45°. An aperture-coupled frequency and polarization selective surface in [16] is reported for dual passband filtering applications. Further, a square loop and a bow-type ring-based center symmetric FSS [17] achieve reflection characteristics in the C-band (4–8 GHz) and X-band (8–12 GHz), with good angular and polarization stabilities. In another study [18], a T-type SRR is placed inside a rectangular SRR to function in the X- and Ku-bands (12–18 GHz) to filter out the unwanted frequencies. However, this FSS illustrates limited polarization and an angular stability up to 45°. An FSS based on vertical and horizontal dipoles with two folded arms shields satellite downlink frequencies [19]. The FSS exhibits a highly stable and polarization-insensitive dual-band (C- and X-bands) frequency response. In [20], a dual-band-notched FSS based on four symmetrically connected square split-ring resonators at the center is reported. It effectively rejects WiMAX and X-band signals, and a passband lies between the two rejection bands.

This study presents a novel low-profile dual-band shield for EMI shielding applications. The FSS unit cell is miniaturized and consists of a modified Jerusalem crossed loop (MJCL) and a corner-modified square loop (CMSL) arranged on a single-layered dielectric. It simultaneously mitigates EMI in the X-band uplink and Ku-band downlink frequencies. It preserves a good angular stability up to ±75° and exhibits a polarization-independent response. The FSS shield is designed and analyzed using a commercially available 3D EM simulator (Ansys HFSS). An ECM is also presented to describe the resonance mechanism. Section 4 reports the measurement setup and measured results of the dual-band-notched FSS filter. Finally, Section 5 concludes the paper.

## 2. Unit Cell Design Configurations

The design layout of the anticipated band-reject FSS unit cell is shown in Figure 1. The unit cell consists of two independent resonant structures, CMSL-FSS and MJCL-FSS, imprinted on a single-layer dielectric. The FSS is designed on a low-loss RT/droid 5880 substrate having a thickness of t = 0.787 mm, ${Ԑ}_{r}$ = 2.2, and tanδ = 0.0009. Thus, the unit cell has the overall dimensions of ${7\times 7\;mm}^{2}$.

### 2.1. Modified Jerusalem Crossed Loop (MJCL) FSS

The MJCL-FSS geometry is principally evolved through a three-step process, as illustrated in Figure 2. Initially, a diagonally placed Jerusalem cross is considered, and a square loop is incorporated at each leg end of it. Further, rectangular slots are etched from the geometry. This helps to miniaturize the unit cell size and achieve a resonant length in a confined space. Finally, the edges at the design center are chamfered at radii of ‘r’ and ‘R’. These radii are further optimized to accomplish a stopband in the Ku-band.

### 2.2. Corner-Modified Square Loop (MJCL) FSS

The CMSL-FSS is depicted in Figure 1b. Primarily, a conventional square loop is selected. Next, the square loop is split up into four segments. Then, rectangular strips are etched from each segment at the corners. In addition, each segment is truncated at the corner to obtain a split-ring resonator structure. Further, each split-ring resonator is joined together to form a truncated square loop element to obtain a miniaturized unit cell, as shown in Figure 2. Thus, the CMSL notches in the X-band. This illustrates better angular stability over the conventional square loop when incorporated with the MJCL-FSS in the co-planner arrangement.

**Figure 1 sensors-24-03333-f001:**
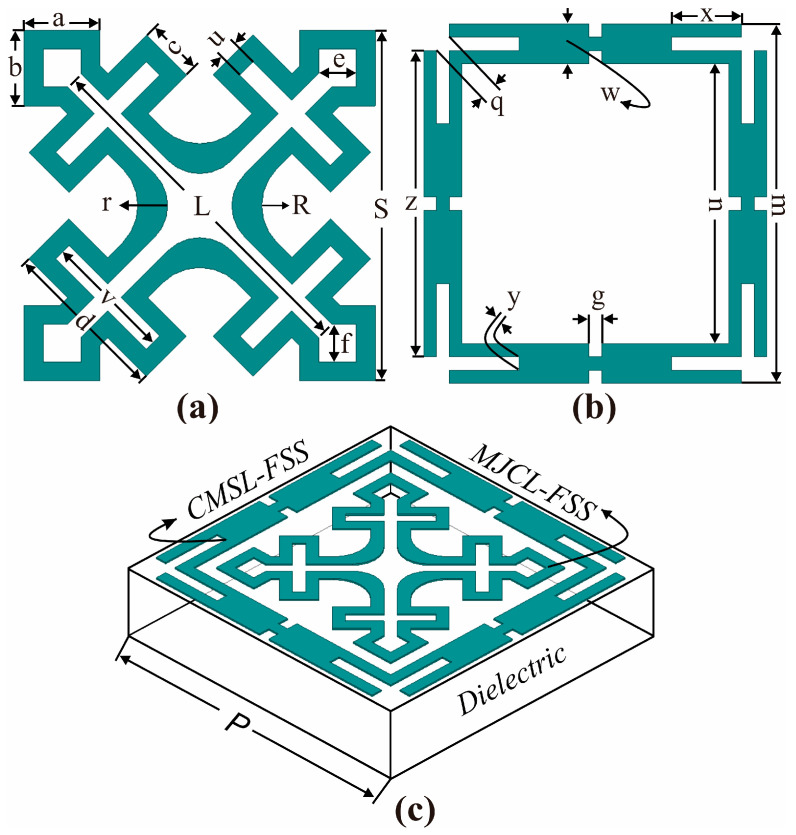
FSS unit cell: (**a**) MJCL-FSS element; (**b**) CMSL-FSS unit element; (**c**) perspective view of the EM shield in co-planner design configuration.

**Figure 2 sensors-24-03333-f002:**
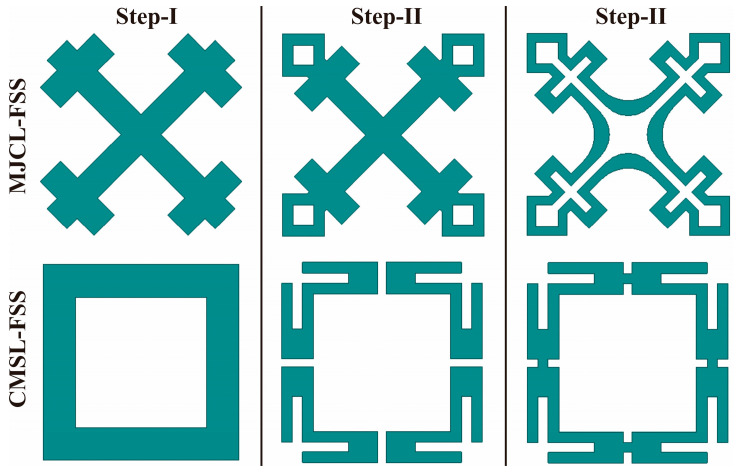
Stepwise design procedure of the elements of the FSS unit cell.

### 2.3. Co-Planner Design Configuration

Figure 1c signifies the co-planner topology of the FSS shield, where the MJCL and CMSL elements are arranged on the top surface of the dielectric. The MJCL is placed inside the CMSL element. It is noticed that in this arrangement, the FSS structures exhibit improved dual-band reject characteristics compared to their singular performances in terms of attenuation, bandwidth, and notch selectivity. However, the first resonance frequency is shifted downwards due to the electromagnetic coupling effect between the resonating elements. As a result, the attenuation, notch selectivity, and stop bandwidth of band II are significantly improved. In this configuration, the CMSL and MJCL resonate at 7.9 GHz, X-band uplink, and 11.9 GHz, Ku-band downlink, frequencies. Thus, the FSS effectively alleviates the unwanted EMI occurring at the X- and Ku-bands. Moreover, Figure 1c shows a 3D view of the EM shield. Table 1 shows optimized values of the design variables of the FSS unit elements.

## 3. Results and Discussions

### 3.1. Simulation of the Proposed Unit Cell

This study uses the commercially available three-dimensional (3D) full-wave FEM-based solver, Ansys HFSS, to design and optimize the anticipated FSS. Initially, a simple Jerusalem cross (JC) structure rotated around the axis at 45° is implemented, which resonates at 20 GHz, as depicted in Figure 3a. Then, in stage II, a square loop is incorporated at the ends of the JC, which increases its electrical length. Finally, rectangular strips are etched to form a loop structure. As a result, the resonant frequency shifts downwards. Thus, the MJCL acts as a bandstop filter at 11.9 GHz in the Ku-band, with an attenuation of 37 dB. Furthermore, in the case of the CMSL-FSS, a conventional square loop (SL) is simulated first, and its frequency response is depicted in Figure 3b. It is noticed that the SL resonates at the 10.5 GHz X-band frequency. Therefore, in step II, the SL is split into four segments and truncated to form a split-ring resonator (SSR) structure. It is observed that the resonance frequency considerably shifted upwards. Finally, these SSRs are joined together to form a compact corner-modified SL. Henceforth, the CMSL-FSS resonates at a significantly lower frequency than the square loop in step I. In addition, the CMSL structure assures more miniaturization and a good angular stability over the conventional SL when used with the MJCL in the co-planner design configuration. Figure 3c presents the frequency response of the FSS structure in the co-planner arrangement.

### 3.2. Shielding Effectiveness

The electromagnetic shielding performance of the FSS unit cell is analyzed in terms of its shielding effectiveness (SE) as a function of frequency for both the TE and TM polarized waves. Thus, the SE is defined as the ratio of the transmitted electric field component to the incident electric field component on the structure and is expressed in Equation (1).
(1)$$SE\;\left(dB\right)=-20\;{log}_{10}\left|\frac{E_{t}}{E_{i}}\right|,$$

Figure 3c reveals the SE of each element of the FSS unit cell individually and together under normal incidence.

### 3.3. Angular Stability Analysis

Angular stability is a phenomenon in which the EM behavior of an FSS surface is rigorously investigated over different oblique incident angles and polarization states. Since the design is four-fold symmetric, perpendicular, and parallel polarizations of the incident plane waves, results in a similar response. Furthermore, angular stability also depends on the unit cell periodicity ‘*P*’ in an FSS array, and to prevent grating lobes, the following relation must be satisfied:(2)$$P<\frac{\lambda_{o}}{1+sin\theta },$$
where $\lambda_{o}$ represents the free-space wavelength, and (*θ*) is the incident angle. Therefore, the value of ‘*P*’ in Equation (2) should be minimal to minimize the angle sensitivity. In this study, for the incident angle (θ=75°), ‘*P*’ should be smaller than 13.2 mm. Theoretically, with the increase in incidence angle, the value of ‘*P*’ is reduced, and the array becomes densely packed, and vice versa. In addition, the relative deviation in frequency response due to angle variations can be calculated using Equation (3).
(3)$$△f=\frac{f_{Z}-f_{oblique}}{f_{Z}},$$
where ∆f, $f_{z},$ and $f_{oblique}$ represent the relative deviation, the transmission zero frequency, and the oblique angle frequency, respectively. Figure 4a specifies the SE of the unit cell at a normal incidence for the TE and TM polarized waves. It is noticed that the FSS exhibits an identical response for TE and TM wave modes. Thus, the unit cell accomplishes SEs of at least 48 dB and 45 dB at the 7.9 GHz and 11.9 GHz frequencies in the X- and Ku-bands. Moreover, the FSS offers a 10 dB bandwidth of more than 2.60 GHz and 2.70 GHz at its operating bands. In addition, Figure 4b,c illustrates the effective shielding characteristics of the FSS as a function of oblique angle under TE and TM polarization up to 75°. The results reveal that the angle variations do not affect the unit cell EM performance. The FSS ensures highly selective and angularly stable spectral responses at its operating bands for the TE and TM polarized waves. However, after 60°, the rejection bandwidth varies within the acceptable limits as the incident angle increases to 75°. Further, it is observed that the notch selectivity also improves over angle variations. The bandwidth and fractional bandwidth can be calculated as follows:(4)$$BW\;\left(GHz\right)=f_{H}-f_{L},$$
(5)$$FBW\left(\%\right)=\frac{BW}{f_{C}}\times 100,$$

In Equations (4) and (5), $f_{H}$, $f_{L}$, and $f_{C}$ represent the higher, lower, and center frequencies, respectively. Based on Equations (4) and (5), the 10 dB fractional bandwidth of the FSS as a function of angle variations for the TE and TM modes is summarized in Table 2. Furthermore, Figure 4d,e reveals the SE curves of the FSS at various polarization angles at a normal incidence. As the FSS structure is four-fold symmetric, thus it manifests identical responses for the TE and TM polarization states up to 90°. Thus, the FSS unit cell exhibits highly selective, angularly insensitive, and similar responses.

### 3.4. Surface Current Density

The EM waves striking the resonant surface induce currents on the surface. The magnitude of these induced surface currents helps us to analyze the resonance mechanism and the influence of each element on FSS performance. The surface current distribution of the FSS unit cell for the TE polarized wave at a normal incidence is presented in Figure 5. In Figure 5a, the strong current concentration present at the modified corners (SRRs) of the outer element and the weak induced current on the inner structure verify that the CMSL-FSS element resonates at the 7.9 GHz, X-band uplink frequency. At the same time, Figure 5b indicates that the MJCL provides an upper stopband centered at the 11.9 GHz downlink frequency in the Ku-band.

### 3.5. Equivalent Circuit Model

An equivalent circuit model (ECM) of the FSS unit cell is derived using an advanced design system (ADS) to further evaluate its performance obtained using a full-wave simulator. Figure 6a reveals the lumped element formation, and Figure 6b shows the derived lumped circuit model of the proposed FSS. The unit cell consists of two independent elements. The outer element is modelled as a series LC resonator represented by $L_{1}$, $C_{1}$, and $C_{2}$, whereas the inner element is modelled as a combination of series and parallel LC resonators with the lumped elements $C_{3}$,${\;C}_{4}$, $L_{2}$, and $L_{3},$ respectively. The metallic/traces are modelled as inductances, while the apertures/slots are modelled as the inter- and intra-element capacitances, and the dielectric substrate is modelled as a transmission line. The characteristic impedance ($Z_{d}$) of the supporting dielectric can be calculated as $Z_{d}=$ $Z_{0}/\sqrt{\varepsilon_{r}},$ where $Z_{o\;}$= 377 Ω is the characteristic impedance of free space on both sides of the FSS, and $\varepsilon_{r}$ is the dielectric constant of the slab. Moreover, the surface impedance of the outer element (CMSL) for the lower band is expressed in Equation (6).
(6)$$Z_{CMSL-FSS}=j\omega L_{1}+\frac{1}{j\omega C_{1}}+\frac{1}{j\omega C_{2}},$$
where $\frac{1}{j\omega C_{1}}$, and $\frac{1}{j\omega C_{2}}$ are the reactances associated with $L_{1}$, $C_{1}$, and $C_{2},$ respectively. By simplifying Equation (6), we obtain
(7)$$Z_{CMSL-FSS}=\frac{\left(j\omega L_{1}\right)\left(\omega^{2}C_{1}C_{2}\right)-\left(j\omega \right)\left(C_{1}+C_{2}\right)}{\omega^{2}C_{1}C_{2}},$$

Similarly, the surface impedance of the MJCL-FSS for the higher band can be calculated using the same procedure as in Equations (6) and (7).
(8)$$Z_{MJCL-FSS}=j\omega L_{2}+\frac{1}{j\omega C_{3}}+\frac{1}{j\omega C_{4}},$$
(9)$$Z_{MJCL-FSS}=\frac{1-\omega^{2}L_{2}C_{3}}{j\omega C_{3}}+\frac{j\omega L_{3}}{1-\omega^{2}L_{3}C_{4}},$$

Multiple simulations are performed in the ADS to obtain the optimized dual-bandstop response of the FSS at the desired frequencies. Figure 6c shows the SE curves of several simulations for various lumped element values. The optimized values of the lumped parameters are tabulated in Table 3. In addition, Figure 6d shows the comparison of the ECM and full-wave simulations. It is observed that the circuit simulations are in good agreement with the full-wave EM simulations.

## 4. Parametric Analysis

Parametric analysis is conducted to evaluate the performance of the dual-band-reject FSS based on the geometrical parameters, including permittivity, the loss tangent, the substrate thickness, and the inter-element spacing. Figure 7a reveals the shielding characteristics of the presented FSS structure as a function of the dielectric constant of the substrate. It is observed from the figure that the rejection bands are shifted downwards with increments in the dielectric constant. These variations can be predicted by the relationship between the electrical length of the resonating element and the permittivity of the employed dielectric given by Equation (10).
(10)$$l\;\propto \;\frac{1}{\sqrt{\varepsilon_{eff}\;}},$$

It is noticed that the attenuation level remains almost the same for both the rejection bands for various values of the dielectric constant. However, the bandwidth varies within the acceptable limits for the FSS design with respect to permittivity variations. Figure 7b shows the behavior of the FSS as a function of the loss tangent of the substrate used to design the FSS structure. The shielding effectiveness of 48 dB and 45 dB at the X- and Ku-bands is observed for the proposed design when tanδ = 0.0009. It is noticed from the figure that the SE decays nearly exponentially as the tanδ varies from 0 to 1. Figure 7c shows the impact of the thickness of employed dielectric on the performance of the FSS structure. Minor variations in the resonance frequency and bandwidth of the upper band of the FSS are observed with the increase in the thickness of the laminate. However, thickness variations do not affect the lower band performance. Furthermore, the physical footprint of the FSS unit cell is the most significant parameter to determine the resonance frequency. Separation among the resonating elements in an FSS array is generally referred to as interelement spacing, or the period. The overall size of a unit cell increases with an increase in the period. Inter-element spacing has a profound effect on the FSS performance. The shielding effectiveness of the unit cell is analyzed by keeping all the other parameters unchanged.

When interelement spacing ($d_{x}=d_{y}$) is varied from 0.25 to 1.0 mm with a step size of 0.25 mm, the size of the dielectric slab increases, increasing the capacitance. An increase in the capacitance value reduces the resonance frequency; therefore, both the rejection bands are shifted upwards, as expected. Figure 7d signifies the effect of inter-element spacing variations on the EM shielding of the unit cell for the TE polarized wave mode at a normal incidence. A minor reduction in the bandwidth and the notch selectivity of the lower band can be observed, whereas the upper band remains intact. Thus, the overall results specified in Figure 7a–d validate that the FSS structure can be tuned to any other frequency of interest by simply altering the design parameters of the unit cell.

## 5. Experimental Setup and Measurements

To verify the simulated EM shielding characteristics of the proposed FSS structure, a finite prototype is fabricated and tested. The fabricated FSS panel consists of 25 × 33 unit cells with the overall dimensions of 243 mm × 188 mm, as shown in Figure 8a. A free-space measurement setup is employed to validate the EM performance of the FSS shield, as illustrated in Figure 8b. The measurements are carried out in a microwave anechoic chamber. The measurement setup consists of a pair of horn antennas placed on both sides of the absorber screen. These antennas are connected to a vector network analyzer (VNA). Prior calibration is carried out without the FSS shield to calibrate the measurement setup. Then, the fabricated FSS panel is placed in between the transmitting and receiving horn antennas ensuring the far-field condition $d\;\geq 2D^{2}/\lambda$, where *D* is the size of the horn antenna, and λ is the wavelength at the lowest working frequency. Moreover, the SE of the FSS is calculated based on the magnitude of the measured $S_{21}$ in dB as follows:(11)$$SE=-\left|S_{21}\right|\;dB,$$

### 5.1. Normal Angle of Incidence

The EM performance of the FSS panel is measured using the above-mentioned setup for different polarizations and incident angles. Figure 9a shows the measured SE of the FSS for TE and TM polarizations at normal incidence. Since the design is four-fold symmetric, it therefore exhibits an identical spectral response for both the vertical and horizontal waves. The results reveal that the FSS achieves SE of more than 35 dB and 40 dB at both operating frequencies. It is observed that the FSS shield exhibits rejection bandwidths of more than 2.10 GHz and 2.40 GHz for the transverse electric, whereas 2.20 GHz and 2.60 GHz for the transverse magnetic modes of operation at the X- and Ku-bands, respectively.

### 5.2. Oblique Angle Incidence

Moreover, the shielding characteristics of the FSS are also measured for various oblique incidence angles up to ±75°, with a step size of 25°. The frequency response of the FSS shield for TE polarization is illustrated in Figure 9b. It is observed that the stop bandwidth and the notch selectivity slightly increase as the incident angle is varied from 0 to 75°. Furthermore, Figure 9c presents the SE versus frequency curves over angle variations for the TM wave mode. The results are almost analogous and stable for TE polarization. However, a minor deviation in the resonance frequencies is also noticed at 75° for the X- and Ku-bands, respectively. Hence, the overall angular performance of the anticipated FSS shield is stable and within acceptable limits at the desired frequency bands. Nevertheless, minor variations in the results are associated with the finite array size, fabrication, and measurement imperfections.

## 6. Comparative Study

This section summarizes a comparison of this work with various published FSS designs. A summary of the reviewed FSSs is presented in Table 4 to show the significance of the proposed FSS structure over other works. Therefore, it is observed from the table that the anticipated EM shield is novel and highly selective, with an angular stability up to ±75° and polarization-insensitive spectral response with compactness as an additional feature.

## 7. Conclusions

In this paper, a novel, compact, and polarization-insensitive electromagnetic shield with dual-stopband characteristics is presented for SATCOM applications. The unit cell of the anticipated FSS design consists of MJCL and CMSL elements printed over a low-profile dielectric. The FSS offers closely spaced dual-band-reject operation, with a minimum band ratio of 1.52. This miniaturized FSS provides at least 35 dB and 40 dB measured suppression at the 7.9 GHz, X-band uplink, and the 11.9 GHz, Ku-band downlink, frequencies, respectively. More importantly, the FSS possesses a higher level of angular stability up to ±75° and polarization insensitivity, owing to its four-fold structural symmetry. In addition, a lumped circuit model of the FSS is devised to further explain its resonance mechanism. A finite prototype of 25 × 33 unit cells is fabricated, and its performance is tested at various oblique incident angles for both the TE and TM polarization states. Thus, the comparison of the simulated and measured results validates that the proposed FSS geometry is a potential candidate for EMI suppression applications, including, but not limited to, SATCOMs, radome designs, microwave filters, antenna reflectors, and many more.

## Figures and Tables

**Figure 3 sensors-24-03333-f003:**
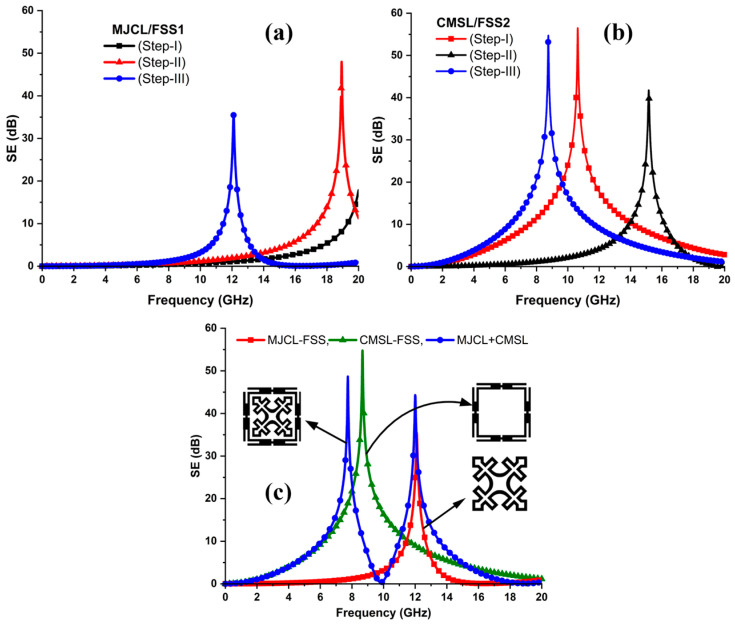
Stepwise performance of FSS elements at normal incidence. (**a**) MJCL. (**b**) CMSL. (**c**) Shielding performances of the MJCL and CMSL individually and in the co-planner configuration.

**Figure 4 sensors-24-03333-f004:**
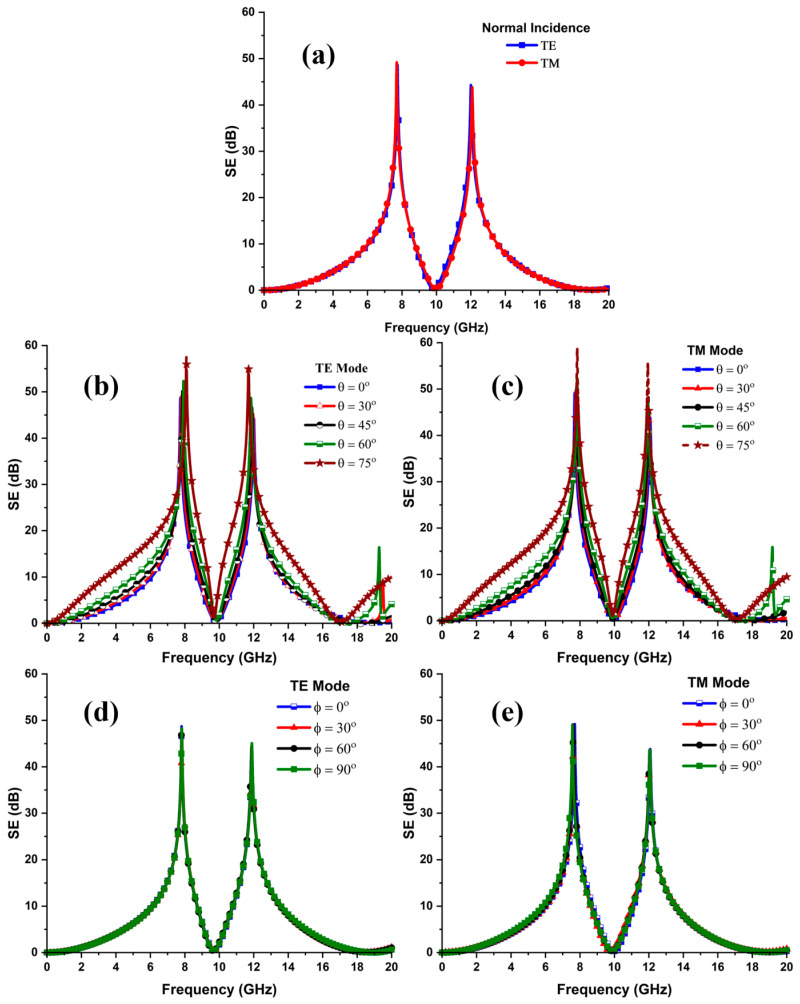
(**a**) Shielding performance of the FSS unit cell at normal incidence for TE and TM polarization states, (**b**) SE of the FSS for TE polarization under oblique angles of incidence ensuring the angular stability up to 75°, (**c**) SE of unit cell structure for TM wave mode, (**d**) FSS response under various polarization angles at normal incidence, confirming polarization insensitivity up to 90° for TE mode, and (**e**) SE plot of the FSS to the polarization angle variations at (θ=0°) for the TM mode.

**Figure 5 sensors-24-03333-f005:**
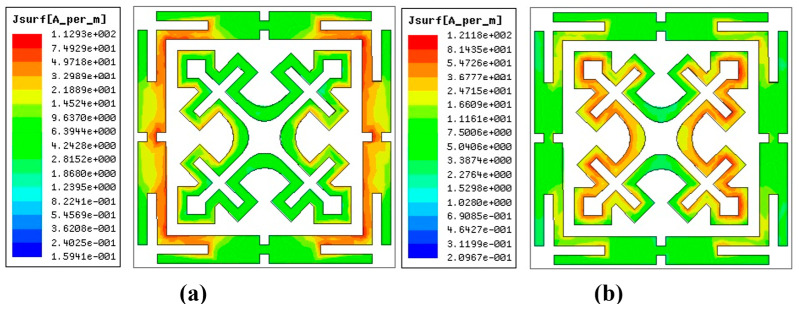
Surface current distribution plot of the FSS unit cell (*θ* = 0°) for TE mode (**a**) at 7.9 GHz and (**b**) at 11.9 GHz.

**Figure 6 sensors-24-03333-f006:**
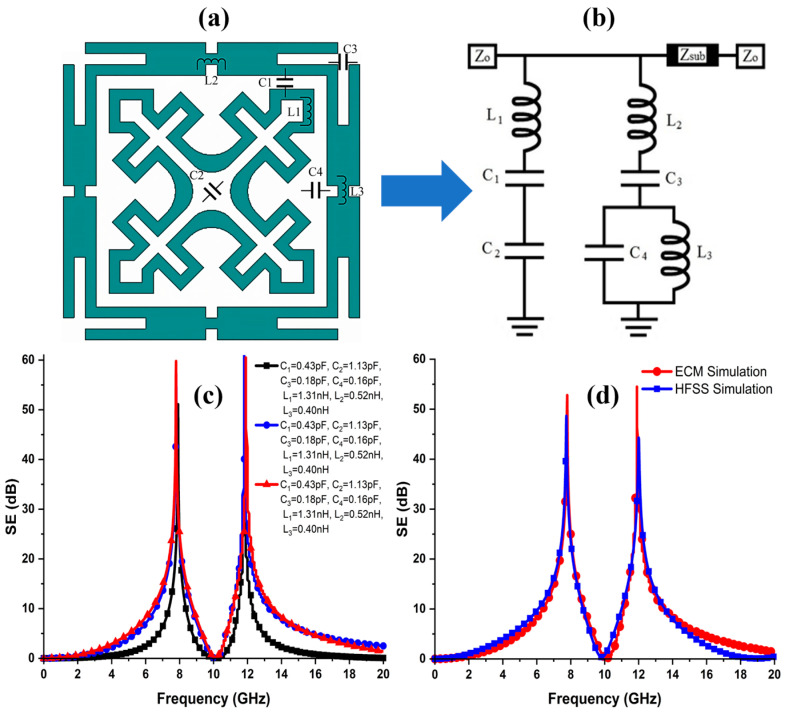
(**a**) Lumped element formation. (**b**) Equivalent lumped circuit model (ECM) of the FSS shield. (**c**) ADS simulations performed to obtain an optimized response. (**d**) Comparison of the ECM and full-wave simulations.

**Figure 7 sensors-24-03333-f007:**
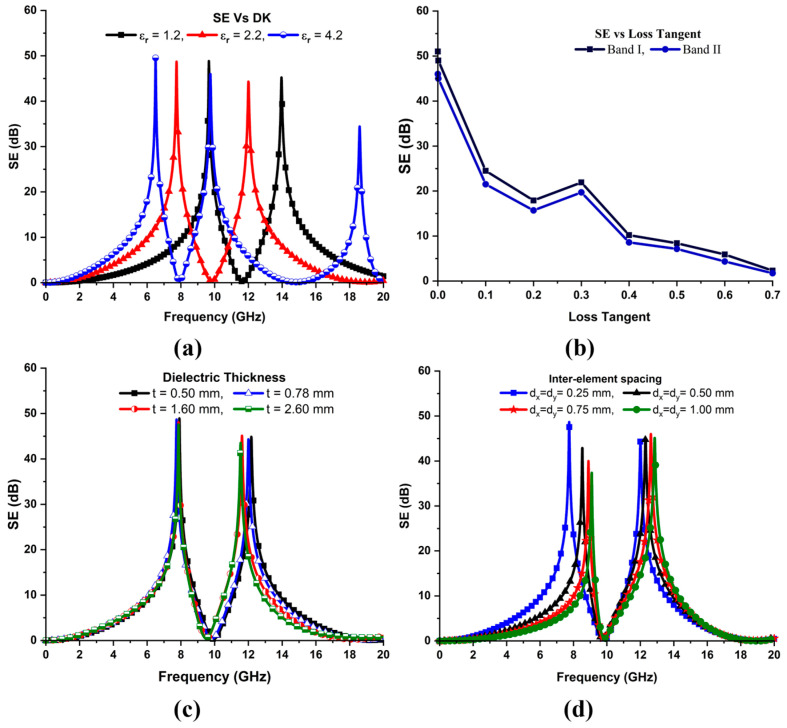
Parametric analysis of the FSS based on its geometrical parameters: (**a**) Impact of dielectric permittivity variations on the performance of the shield. (**b**) SE of the FSS as a function of loss-tangent of the employed substrate. (**c**) Variation in stopbands with respect to the thickness of the substrate of the FSS unit cell at normal incidence. (**d**) Resonant frequency variations with respect to the inter-element spacing of elements at the normal wave incidence TE_Z_-*θ*°.

**Figure 8 sensors-24-03333-f008:**
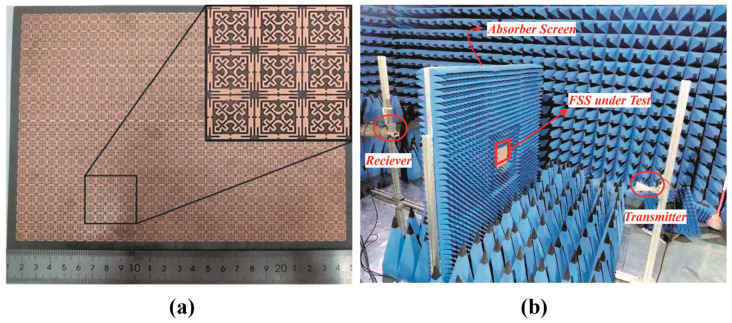
(**a**) Fabricated FSS panel with zoomed view given as inset. (**b**) A photograph of measurement setup inside an anechoic chamber.

**Figure 9 sensors-24-03333-f009:**
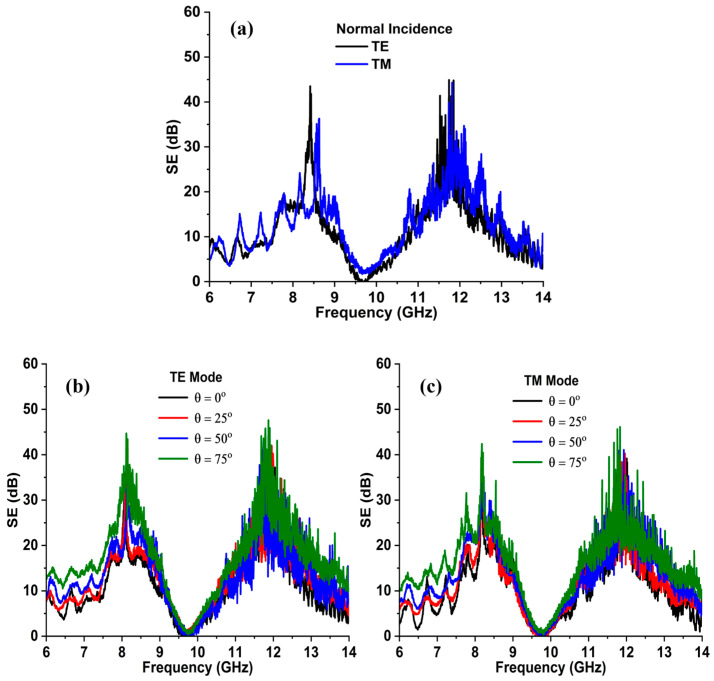
(**a**) Performance of the FSS shield at a normal angle of incidence for the TE and TM polarization states. (**b**) SE versus frequency plot as a function of oblique angle for the TE polarization mode. (**c**) SE performance of the FSS for TM mode under various angles of incidence, ensuring angular stability.

**Table 1 sensors-24-03333-t001:** Design parameters of FSS unit elements.

MJCL-FSS	Parameter	a = b	c	d	e = f	r	R	u	v	S	L
	Values	1.00	0.75	2.50	0.50	1.00	0.90	0.25	2.00	4.65	4.70
CMSL-FSS	Parameter	g	m	n	q	x	y	z	P	-	-
	Values	0.50	6.75	5.25	0.35	1.35	0.50	5.75	7	-	-

**Table 2 sensors-24-03333-t002:** Fractional bandwidth as a function of incidence angle.

Angle (*θ*)	Resonant Frequency (GHz)				Attenuation (dB)				Fractional Bandwidth (%)			
	Band I		Band II		Band I		Band II		Band I		Band II	
	TE	TM	TE	TM	TE	TM	TE	TM	TE	TM	TE	TM
0°	7.90	7.70	11.94	12.08	48.72	45.02	49.21	43.82	35.27	34.76	21.62	20.37
30°	7.91	7.76	11.92	12.01	48.99	45.22	49.65	44.39	39.10	40.45	22.45	21.63
45°	7.95	7.79	11.90	12.00	50.09	46.62	50.21	45.65	45.01	46.25	23.77	23.44
60°	8.02	7.84	11.84	11.94	52.30	48.63	52.32	47.89	58.56	60.75	28.22	27.90
75°	8.09	7.84	11.70	11.94	57.47	55.9	56.63	55.45	92.91	98.12	39.29	36.67

**Table 3 sensors-24-03333-t003:** Optimized lumped parameters.

Equivalent Circuit Values							
$L_{1}$	1.31 nH	$L_{2}$	0.52 nH	$L_{3}$	0.40 nH	-	-
$C_{1}$	1.13 pF	$C_{2}$	0.16 pF	$C_{3}$	0.18 pF	$C_{4}$	0.16 pF

**Table 4 sensors-24-03333-t004:** Comparison with dual-band FSSs reported in the literature for EM shielding applications.

Ref. No.	Unit Cell Size (mm)	Employed Substrate	Operation Band	Angular Stability	Frequency Ratio
[12]	0.087λ × 0.087λ	FR-4	UWB	0–60°	N/A
[13]	0.194λ × 0.194λ	Polyethylene terephthalate	GSM	0–60°	2.11
[14]	0.163λ × 0.163λ	Glass	WiFi/WLAN	0–45°	2.24
[15]	0.21λ × 0.21λ	Paper and Polymer	WiFi/WLAN	0–45°	2.22
[16]	0.161λ × 0.161λ	Taconic RF-35	X	0–50°	N/A
[17]	0.167λ × 0.167λ	FR-4	C/X	0–60°	1.32
[18]	0.26λ × 0.26λ	RO4350B	X/Ku	0–45°	1.91
[19]	0.149λ × 0.149λ	FR-4	C/X	0–60°	1.87
[20]	0.082λ × 0.082λ	FR-4	WiMAX/X	0–70°	3.3
This Work	0.184λ × 0.184λ	RT/duroid 5880	X/Ku	0–75°	1.52

The dimensions of the FSS unit cell are calculated based on wavelength at first resonant frequency.

## Data Availability

Data are contain within article.

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
