# Peer review of "A Dual-Band Polarization-Insensitive Frequency Selective Surface for Electromagnetic Shielding Applications"

_sensors, 2024, doi:10.3390/s24113333_

Round 1

Reviewer 1 Report

Comments and Suggestions for Authors

The manuscript presents a dual-band FSS-based EM shielding structure with polarization-independent property. The manuscript is well-written and the design is verified with simulation and measurements. This reviewer's comments are as follows. 

1. Please explain why the frequency-selective shielding is required rather than shielding over a wide frequency range (e.g. 6-14 GHz).

 2. In the same line, please explain the difference between a mesh metal (for low weight) screen of the same thickness (0.787 mm) and the proposed shielding structure in the performance of shielding. A metal screen might be better than the proposed structure for shielding at 6-14 GHz.

 3. Proposed structure shows SE > 40 dB almost at a single frequency (extremely narrow bandwidth). Even the bandwidth for SE > 20 dB is too narrow. Please explain the use of such a narrow-band shielding structure.

 4. Please explain if 5 layers (for example) of the proposed structure (total thickness of about 4 mm) will increase SE five times.

Reviewer 2 Report

Comments and Suggestions for Authors

In this study, the authors propose an electromagnetic shield with dual stopband characteristics tailored for SATCOM applications. While the research fits well within the journal's scope, and both simulation and experimental results show promise, there are several notable issues that require attention (minor revisions):

1.    Include the frequency ranges of C-, X-, and Ku-bands in the introduction section.

2.    Correct the figure numbering so that Figure 1 is properly referenced in the text (There are two figure 2 in the text).

3.    Elaborate on the mechanism behind the improved dual-band reject characteristics observed in FSS structures, as mentioned in lines 98-99.

4.    Provide a more detailed clarification of the results obtained from Figure 3(c).

5.    Improve the references in the introduction section by incorporating more recent related works. Additionally, cite references to offer insight into alternative approaches to multiple band (https://doi.org/10.1007/s11468-024-02219-2) and broadband selective metasurfaces (Phys. Scr. 99 (2024) 055905) in the introduction section.

6.    Lastly, meticulously review the manuscript to rectify any typos and grammatical errors.

Reviewer 3 Report

Comments and Suggestions for Authors

- It is not clear what the originality of the proposed structure? There are currently many FSS are available.

- Too brief an introduction that does not give an idea of ​​the state of art in this research area.

- The purpose of the equivalent circuit model of the FSS unit cell is not clear. What does it give new, besides the fact that the results of circuit modeling coincide with EM modeling?

- The experiment  section requires comments on how the SE was determined based on the measured S-parameters.

- In the comparison table, if structures operating in too different frequency ranges are compared, then the size more correctly should be given in wavelengths

Round 2

Reviewer 3 Report

Comments and Suggestions for Authors

No objections